# Winter Geometrid Moths in Oak Forests: Is Monitoring a Single Species Reliable to Predict Defoliation Risk?

**Lenka Sarvašová [1]**, **Ján Kulfan [1,\*]**, **Miroslav Saniga [1]**, **Milan Zúbrik [2]** and **Peter Zach [1]**

[1]  Institute of Forest Ecology, Slovak Academy of Sciences, Ľ. Štúra 2, 960 53 Zvolen, Slovakia;
    sarvasova@ife.sk (L.S.); miro.saniga@gmail.com (M.S.); zach@ife.sk (P.Z.)

[2]  National Forest Centre, Forest Protection Service Centre, Lesnícka 11, 969 01 Banská Štiavnica, Slovakia;
    milan.zubrik@nlcsk.org

\*  Correspondence: kulfan@ife.sk

**Abstract:** Species within the group of winter moths (Geometridae) are important oak defoliators in European forests. Adults of these species emerge either in late autumn ('autumn species') or in early spring ('spring species'), and caterpillars of both 'autumn' and 'spring' species appear in spring. The abundance of adults assessed by regular monitoring allows the prediction of the defoliation intensity in trees by caterpillars in the following spring. 'Autumn species' (mostly a single one, *Operophtera brumata*) are monitored by forestry practices as pests, whereas 'spring species' are often not paid any attention. We hypothesised that 'spring species' could also have an important share in caterpillar assemblages in oak forests. We aimed to study the proportions between 'autumn' and 'spring' species in adult and larval stages. In a xeric thermophilous oak forest in southern Slovakia, Central Europe, we collected adult moths using sticky bands set up on trunks of *Quercus pubescens* during the winter of 2014–2015 and caterpillars from other trees belonging to the same oak species over the following spring. We also captured caterpillars from several oak species in various areas and over different years in Slovakia and Bulgaria, and we compiled relevant literature data. 'Spring species' recorded from a unique forest as adults during winter and as caterpillars in the following spring were significantly more abundant than 'autumn species.' Moreover, 'spring species' from Slovakian forests, regardless of the locality, the oak species and the year of sampling, reached high proportions, mostly over 50% of individuals in caterpillar assemblages. The 'spring species' *Agriopis leucophaearia* was the most abundant, followed by the 'autumn species' *Operophtera brumata.* 'Spring species' accounted for more than 50% of individuals in caterpillar assemblages in the Balkan Peninsula (Bulgaria) concerning one case, and they were of little importance in northern Greece. We recommend monitoring all winter moth adults ('autumn' and 'spring' species together) continuously in forestry practices, using sticky bands on oak trees from late autumn to early spring.

**Keywords:** forest protection; geometridae; caterpillars; *Quercus*; *Agriopis leucophaearia*

## 1. Introduction

Caterpillars occurring in the spring are important tree defoliators in temperate forests [1–4]. In spring, caterpillar assemblages, some geometrid species of economic importance, belong to a so-called ecological group of winter moths [5]. Adults from this group do not emerge during the growing season, i.e., either in late autumn ('autumn species' for this study) or in early spring ('spring species'). Beside this, females have reduced wings (brachypterous/apterous females) [5].

*Operophtera brumata* (Linnaeus, 1758) and *Erannis defoliaria* (Clerck, 1759) are generally considered to be the most important pests within the group of winter moths in Central Europe [1,6–8]. They

occur very frequently and cause extensive defoliation in oak forests within Europe as well as in North America [9–11].

To monitor the population density of *O. brumata* and *E. defoliaria* belonging to 'autumn species' and to predict the damage to foliage caused by their larvae in the following spring, sticky bands or glue strips placed on tree trunks are often used [12–19]. Such monitoring of these particular species is usually set up for late autumn and early winter, i.e., when adults occur. Such timing does not allow the monitoring of winter moth adults emerging in late winter and early spring, i.e., 'spring species'. Caterpillars of these 'spring species' co-occur with those of *O. brumata* and *E. defoliaria*, causing damage to leaves during spring. In European deciduous forests, the complex of 'spring species' consists of species such as *Alsophila aescularia* (Denis and Schiffermüller, 1775), *Agriopis leucophaearia* (Denis and Schiffermüller, 1775), *Agriopis marginaria* (Fabricius, 1776), *Apocheima hispidaria* (Denis and Schiffermüller, 1775), and *Phigalia pilosaria* (Denis and Schiffermüller, 1775).

In this document, we addressed (1) the 'spring species' abundance of winter moths in Europe's temperate oak forests in comparison to the abundance of 'autumn species' and (2) whether monitoring winter moth adults in late autumn and early winter (October–December) is an adequate method to assess the overall significance (performance) of their larvae as pests in oak forests.

We hypothesised that 'spring species' could reach large proportions in caterpillar assemblages of winter moths on oaks, judged from their high abundance in some regions and years [1,20–26] and, therefore, they could be paid greater attention by being included in programmes of forest pest monitoring.

## 2. Materials and Methods

### 2.1. Collection of Moths

We collected adult moths according to a method [27] using sticky bands in the Krupinská planina plateau (southern Slovakia, Central Europe; 48°10′0.19″ N, 18°59′46.08″ E) from November 2014 to April 2015. We set up sticky bands for moths on the trunks of 45 randomly selected mature trees belonging to pubescent oak (*Quercus pubescens* Willd.) in a xeric thermophilous oak forest dominated by pubescent oak and Turkey oak (*Quercus cerris* L.). The study area was approximately 50 ha. We regularly controlled these bands and replaced them with fresh ones to avoid their saturation by moth males. Captured moths were identified to the species level [28] and counted in the laboratory.

Over the following spring, in early May 2015, we captured caterpillars from pubescent oaks in the same forest area by the method known as beating. We knocked them off from the lower branches of mature trees. These trees were different from the trees used for obtaining adults during the previous winter. We used a stick and a circular beating tray (diameter of 1.0 m) to obtain caterpillars. The heights for sampling varied from 1.0 to 3.0 m, measured from the ground. A total of 23 trees were sampled. One sample constituted caterpillars that were collected from three branches (each branch 1 m long) from each tree. The larvae were preserved in 70% ethanol and identified in the laboratory [3,29]. From both winter and spring samples, only individuals belonging to winter moth geometrids were selected and separated into 'autumn' and 'spring' species.

To obtain wider knowledge about proportion between numerous caterpillars of 'autumn' and 'spring' geometrids, we also collected winter moth larvae from several oak species in various areas, as well as over years in Slovakia and Bulgaria (for more details, see Samples S08–S12 and S15, Table 1). We further analysed relevant data from the literature [25,30–32]. The available data refer to caterpillars on *Q. cerris*, *Q. pubescens*, *Quercus frainetto* Ten., *Quercus dalechampii* Ten., *Quercus polycarpa* Schur and a few other oak species in Slovakia, Bulgaria and Greece. Caterpillars were captured from trees, mostly by the beating method [30]; samples were obtained by hand collecting. For more details, see Table 1.

**Table 1.** Survey of sampling caterpillar assemblages on oaks in Slovakia (SVK), Bulgaria (BGR) and Greece (GRC).

| Sample | Country | Area | Coordinates | Oak Species | Year | Sample | Data Source |
|---|---|---|---|---|---|---|---|
| S01 | W SVK | Malé Karpaty Mountains | 48°19′ N 17°17′ E | *Q. dalechampii* | 2000–2002 | 150 branches | Kulfan (2012) |
| S02 | W SVK | Malé Karpaty Mountains | 48°22′ N 17°19′ E | *Q. dalechampii* | 2000–2002 | 150 branches | Kulfan (2012) |
| S03 | W SVK | Malé Karpaty Mountains | 48°29′ N 17°23′ E | *Q. dalechampii* | 2000–2002 | 225 branches | Kulfan (2012) |
| S04 | W SVK | Malé Karpaty Mountains | 48°32′ N 17°31′ E | *Q. dalechampii* | 2000–2002 | 150 branches | Kulfan (2012) |
| S05 | W SVK | Malé Karpaty Mountains | 48°32′ N 17°31′ E | *Q. cerris* | 2000–2002 | 225 branches | Kulfan (2012) |
| S06 | W SVK | Malé Karpaty Mountains | 48°44′ N 17°46′ E | *Q. polycarpa* | 2011 | 340 branches | Parák et al. (2012) |
| S07 | W SVK | Malé Karpaty Mountains | 48°44′ N 17°46′ E | *Q. pubescens* | 2011 | 140 branches | Parák et al. (2012) |
| S08 | SW SVK | Podunajská pahorkatina upland | 48°12′ N 18°24′ E | *Q. cerris, Q. petraea* | 2014 | 100 branches | original data |
| S09 | SW SVK | Podunajská pahorkatina upland | 48°12′ N 18°24′ E | *Q. cerris, Q. petraea* | 2015 | 100 branches | original data |
| S10 | S SVK | Krupinská planina plateau | 48°10′ N 18°59′ E | *Q. cerris* (adult trees) | 2015 | 102 branches | original data |
| S11 | S SVK | Krupinská planina plateau | 48°10′ N 18°59′ E | *Q. cerris* (young trees) | 2016 | 60 branches | original data |
| S12 | S SVK | Krupinská planina plateau | 48°10′ N 18°59′ E | *Q. pubescens* (young trees) | 2016 | 60 branches | original data |
| S13 | N GRC | Mount Holomontas | 40°25′ N 23°30′ E | *Q. dalechampii, Q. frainetto, Q. trojana, Q. pubescens, Q. pedunculiflora* | 1997–1998 | ≈ 710 branches | Kalapanida and Petrakis (2012) |
| S14 | BGR (N, S, E, W, Central) | | 42°03′ N 24°54′ E | *Q. cerris, Q. frainetto, Q. petraea, Q. robur, Q. pubescens* | 2009–2011 | 300 oaks | Georgieva et al. (2014) |
| S15 | S BGR | Rhodope Mountains | 41°20′ N 25°22′ E | *Q. frainetto* | 2017 | 540 branches | original data |

### 2.2. Statistical Data Analysis

Generalized estimating equations (GEE) [33] were applied to make inferences about populations of winter moths developing on oak trees. We estimated and compared abundances of adult moths belonging to 'autumn' and 'spring' species (males and females), in particular from the two predominant species, *Operophtera brumata* and *Agriopis leucophaearia*, from females of 'autumn' and 'spring' species as well as from females of *O. brumata* and *A. leucophaearia* found trapped by bands on oak trunks. The GEE approach was further applied to detect differences in larval abundances of 'autumn' and 'spring' species and in that of *O. brumata* and *A. leucophaearia* on oak branches. The response variable was the moth abundance and the explanatory variable included the factor 'species group' ('autumn' or 'spring') or the factor 'species', which included the names of the two predominant species (*O. brumata, A. leucophaearia*) as specific factor levels. As moth adults were collected on different perimeters of trunks, an offset value (the natural logarithm of trunk perimeter) was used to account for this variation. The grouping structure for the GEE was defined by the specific block identifier (tree id), accounting for potential within-tree correlation (exchangeable correlation structure) between abundances of 'autumn' and 'spring' species and/or that of *O. brumata* and *A. leucophaearia,* respectively. Trees with a large perimeter could host more individuals of 'autumn' as well as 'spring' species than those with a smaller perimeter.

Statistical analyses were performed in the R free software environment for statistical computing and graphics [34], using the 'geepack' package [35]. Some of the data such as mean abundances of adults, females and larvae belonging to 'autumn' and 'spring' species were presented by bar plots at the original scale (after exponentiation of the natural logarithm for mean values from GEE), and 95% confidence intervals for the estimated means were drawn.

## 3. Results

### 3.1. Assemblages of Adult Moths

Using sticky bands, a total of 21,692 winter moth adults were collected from November 2014 to April 2015. All individuals (males and females) of 'spring species' were significantly more abundant than those of 'autumn species' (Figure 1), and the same held for females (Table 2). The total number of individuals belonging to 'spring species' was 2.6 times higher in comparison to that of 'autumn species'.

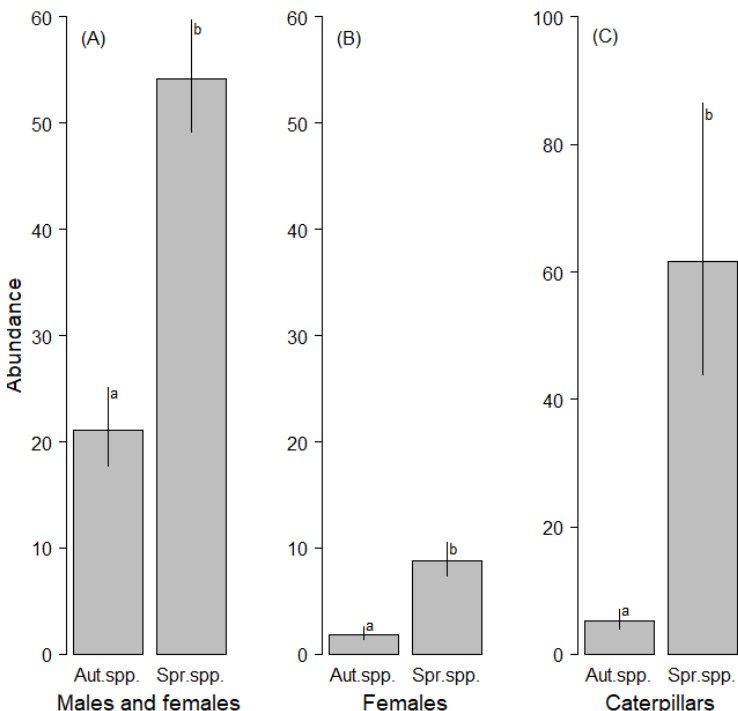

**Figure 1.** Abundance of winter geometrid moths (adults and caterpillars) on oak trees in Central Europe (Slovakia). (**A**) Males and females trapped per 1 dm of trunk perimeter at breast height (nc = 45, cs = 2); (**B**) females stuck per 1 dm of trunk perimeter at breast height (nc = 45, cs = 2); and (**C**) caterpillars per 1 m branches in length (nc = 23, cs = 2). Bars denote the mean abundance of moths or caterpillars; vertical lines show the 95% confidence intervals (CIs) for the means. Different letters above the CIs indicate significant difference. Aut. spp., 'autumn species'; Spr. spp., 'spring species'; nc, number of clusters; cs, cluster size.

Seven species of winter moths were recorded in total. The 'spring species' *A. leucophaearia* comprised a majority of captured moths (71% of all individuals and 81% of females). The 'autumn species' *O. brumata* was the second most abundant, accounting for 21% of all individuals and 12% of females belonging to winter moths. *Agriopis leucophaearia* was significantly more abundant than *O. brumata* (Table 2). The less abundant 'spring species' were *A. marginaria*, *A. hispidaria* and *P. pilosaria*. As for 'autumn species', they were *Erannis defoliaria* and *Alsophila aceraria* (Denis and Schiffermüller, 1775).

Females were less abundant than males on sticky bands and comprised 8% of all captured individuals belonging to 'autumn species' and 16% of 'spring species'. The proportion of females was 8% for *O. brumata* and 16% for *A. leucophaearia*.

**Table 2.** Results of paired comparisons between moth groups or moth species.

| Moth Group/Species | Wald Statistics | Stat. Significance | Corel. Parameter $\alpha$ |
|---|---|---|---|
| 'spring species' (males + females) vs. 'autumn species' (males + females) | 98.7 | $p < 0.0001$ | 0.18 |
| 'spring species' (females) vs. 'autumn species' (caterpillars) | 72.9 | $p < 0.0001$ | 0.02 |
| *Agriopis leucophaearia* (males + females) vs. *Operophtera brumata* (males + females) | 178.0 | $p < 0.0001$ | 0.20 |
| *Agriopis leucophaearia* (females) vs. *Operophtera brumata* (females) | 101.4 | $p < 0.0001$ | 0.02 |
| 'spring species' (caterpillars) vs. 'autumn species' (caterpillars) | 140.3 | $p < 0.0001$ | 0.07 |
| *Agriopis leucophaearia* (caterpillars) vs. *Operophtera brumata* (caterpillars) | 136.5 | $p < 0.0001$ | 0.07 |

*3.2. Assemblages of Caterpillars*

Using the beating method, in May 2015, we sampled from oak branches a total of 4618 caterpillars belonging to the winter moths group. As in the adults captured during the previous winter, the abundance of caterpillars belonging to 'spring species' was significantly higher than that of 'autumn species' (Figure 1, Table 2). The total number of caterpillars belonging to all 'spring species' was 11.6 times higher than that of all 'autumn species'. In addition, the most abundant species were the same—*A. leucophaearia* and *O. brumata*. *Agriopis leucophaearia* was significantly more abundant than *O. brumata* (Table 2). *Agriopis leucophaearia* caterpillars accounted for 89% of all winter moths and *O. brumata* caterpillars for 6%. Other species of winter moths such as *E. defoliaria*, *A. aceraria*, *A. aescularia*, *A. marginaria*, *A. hispidaria*, *P. pilosaria* and *Lycia pomonaria* (Hübner, 1790) were not abundant.

The majority of observed caterpillars were green (71%). The ones belonging to the most abundant species, *A. leucophaearia*, occurred in two colours—green and a darker form (brown-spotted to dark brown) (73% and 27% of specimens recorded, respectively). These two forms are well known [29,36]. Green caterpillars of this species predominated markedly (91%), followed by *O. brumata* (9%) and other species (*A. aescularia* and *A. aceraria*, both <1%).

In Slovakia, 'spring species' comprised a considerable part of caterpillar assemblages regardless of the oak species, the area and the year (Figure 2). Here, in several cases, *A. leucophaearia* obviously predominated. In the Balkan Peninsula, caterpillar assemblages consisted mainly of 'autumn species' (Figure 3). *Agriopis leucophaearia* did not occur in high numbers in most localities except for one location in southern Bulgaria (Rhodope Mountains), where it accounted for over 40% of all species on *Q. frainetto* (Figure 3).

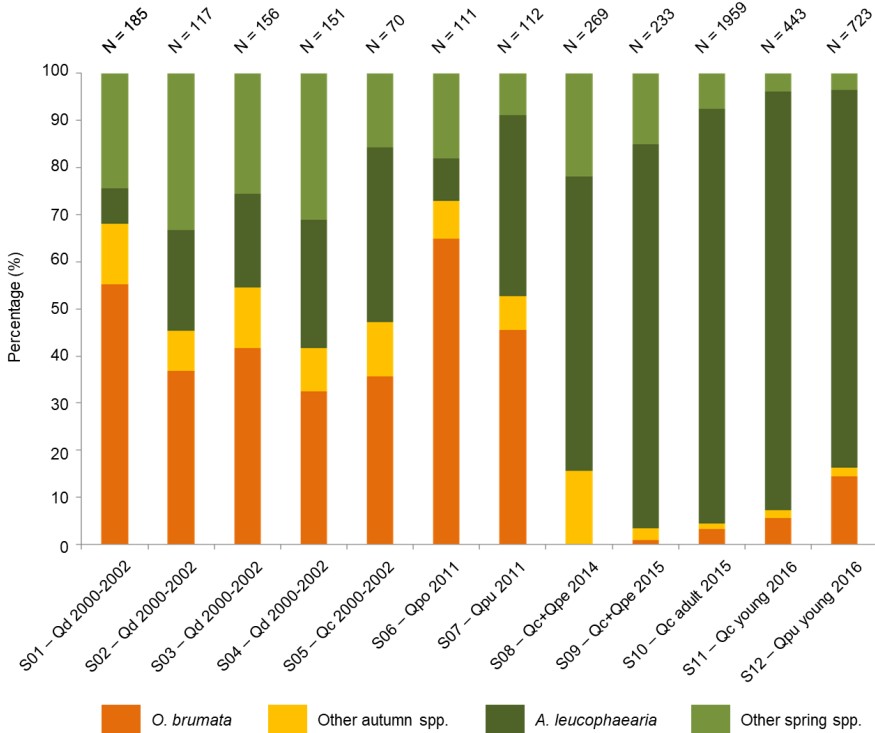

**Figure 2.** Composition of caterpillar assemblages consisted of winter moth species on oaks in various areas in Slovakia. S01–S12: sample number (for details, see Table 1); Qd, *Quercus dalechampii*; Qc, *Q. cerris*; Qpe, *Q. petraea*; Qpo, *Q. polycarpa*; Qpu, *Q. pubescens*.

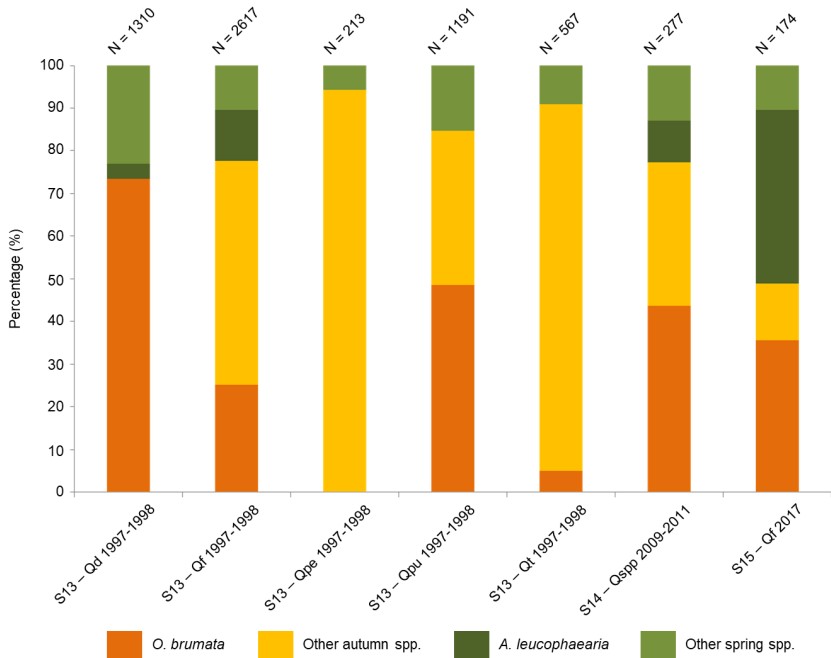

**Figure 3.** Composition of caterpillar assemblages consisted of winter moth species on oaks in various areas in the Balkan Peninsula. S13–S15: sample number (for details, see Table 1); Qd, *Quercus dalechampii*; Qf, *Q. frainetto*; Qpe, *Q. petraea*; Qpu, *Q. pubescens*; Qt, *Q. trojana*; Qspp includes the following *Quercus* species: *Q. cerris*, *Q. frainetto*, *Q. petraea*, *Q. pubescens*, *Q. robur L.*, and *Q. pedunculiflora*.

## 4. Discussion

During our field studies in 2014–2015, we recorded seven species of winter geometrid moths in the adults and nine species in the caterpillars. They are all widely spread within Europe and most of them occur frequently in Central European oak forests [3,28,37]. Concerning all individuals, 'spring species' clearly dominated over 'autumn species' for both adults and larvae. In both cases, the 'spring species' *A. leucophaearia* occurred more abundantly than the 'autumn species' *O. brumata*. The abundance ratio between these two species in adult and caterpillar assemblages was different. This could have resulted from the various methods used to collect adults and caterpillars. Adults were continuously collected on tree trunks over a longer period, whereas caterpillars were collected on lower branches over a single sampling occasion.

Data about the composition of caterpillar assemblages on oak trees in various areas within regions of Europe enable us to consider a broader relevance for our results. The ratio between 'spring' and 'autumn' species within the group of winter geometrid moths on various oak species and from different regions and over years was not identical. However, 'spring species' in assemblages from forests in Slovakia, regardless of the locality, the oak species and the year of sampling, were shown to reach high proportions, in most cases, of over 50% of the specimens recorded (Figure 2). In general, there is a lack of knowledge about the composition of caterpillar assemblages on oak trees in Europe. 'Spring species' are suggested to be abundant and, hence, important in some countries from Western and Central Europe, e.g., the British Islands [38], Germany [20], Austria [39], Poland [40], Slovakia [3,26,32] and also Hungary [41,42]. The importance of 'autumn' geometrid species is known, mainly thanks to the frequently studied *O. brumata*, which has a large outbreak distribution within Europe [16]. Data about outbreaks or high numbers of other 'autumn species', such as *E. defoliaria*, are infrequent [4,11,20,23,39,43,44].

A smaller proportion of 'spring species' in caterpillar assemblages has been reported from south-eastern Europe, although our samples from southern Bulgaria indicate their occurrence in high numbers. It seems that, in the Balkan Peninsula, these species may be important [4,23,45]. In the Mediterranean area, their importance as pests decreases due to small numbers [16,24,30,46–48].

Damage to foliage of deciduous forest trees is often caused by several co-occurring species [3,39,42]. However, questions about the synchronisation of their population dynamics, or their outbreaks, have not been answered [49]. Many studies have proposed that particular geometrids, in agreement with our results, are not synchronised with each other (or with the population cycle of the well-known species, *O. brumata*) [50–53]. Other research has shown similar dynamics of certain species [42, 54]. The comparisons regarding the abundances of *A. leucophaearia* and *O. brumata* in our work suggest possible differences between the population cycle of *A. leucophaearia* and that of *O. brumata*. Consequently, we suppose that outbreaks of particular brachypterous winter geometrid moths may not always be expected along with those of the frequently monitored *O. brumata*.

In forestry practices, the sticky bands attached to tree trunks are targeted to record females of 'autumn species' (mostly *O. brumata*) to estimate and predict a defoliation risk in forests over the following spring [14,17,55–57]. The critical value for defoliation risk is 0.8–1.0 female per 1 cm of the tree trunk's circumference in the case of *O. brumata* and 0.3–0.4 female per 1 cm of the tree trunk's circumference for *E. defoliaria* [17,58]. Another method to monitor winter moth adults in forestry practices, namely pheromone trapping, also focuses primarily on 'autumn species', mostly *O. brumata* [59–61]. As 'spring species' may predominate, monitoring winter geometrid moths from October to December is inaccurate. Therefore, the recording of adults should be performed from late autumn to early spring. This approach can even improve data about 'autumn species'. In southern Europe and the British Islands, for example, the occurrence of adults belonging to several 'autumn species', including *O. brumata*, is prolonged up to January, February or even March [47,62–65]. The phenology of winter moths depends on temperature [66]; the weather influences the timing of adult emergence in particular years. Climate warming could shift the flight period of *O. brumata* in Central Europe towards mid- or late winter, resembling the situation in southern Europe [65].

In almost all species captured on sticky bands, the number of brachypterous females was smaller than that of winged males. This is customary when using this collecting method [67,68]. The number of individuals collected with these bands depends on both the abundance and activity of the moths. Winged males are more active than brachypterous females, and consequently, they come into contact with sticky bands more frequently. In addition, these females captured could attract males using pheromones [69–74]. Counting inconspicuous females, performed by routine forestry practices, is relatively difficult. Counting winged males or males and females together could provide a more appropriate method to assess moth abundance. Even if male bodies are removed from sticky bands by predators (birds, small mammals), their wings usually remain where they were glued. In the case of females, when eaten by predators, only their legs stay on these bands, often overlaid by males or their wings, so it is difficult to notice and count them. In addition, males are easier to identify into species.

Field identification of phyllophagous caterpillars into geometrid species is not easy. In general, they are green or dark [28,29,36]. Our samples from *Q. pubescens* obtained in the spring of 2015 suggest that green caterpillars could predominate. Almost three-quarters of the collected caterpillars were green; more than 90% of them were identified as *A. leucophaearia*, followed by *O. brumata*, *A. aescularia* and *A. aceraria*. It is difficult to distinguish *O. brumata* (the most "popular" winter moth species) from other green caterpillars, especially when they are young instars. The misidentification may cause an overestimation of *O. brumata* and an underestimation of other species (e.g., *A. leucophaearia*).

## 5. Conclusions

We may conclude that monitoring winter moth adults conducted only in late autumn and/or early winter is not reliable to reflect their overall abundance and to predict the defoliation risk caused by their caterpillars in the following spring. We recommend setting up sticky bands on trees from late autumn to early spring and to look at them continuously. Monitoring males or males and females together may be easier and more accurate than monitoring females only. There is a high risk of misidentification for winter moths (especially females and caterpillars) in routine field monitoring using forestry practices. We propose the monitoring and/or checking of this important group of moths as a whole, rather than focusing only on certain abundant and well-known species. Certain lesser known species such as *A. leucophaearia* appear to be at least as important defoliators as *O. brumata.* They, therefore, should be paid greater attention in the temperate oak forests within Europe. Further research is needed to assess the critical number of winter geometrid moths with regards to defoliation risk.

**Author Contributions:** L.S. analysed and collected data, carried out literature searches and co-wrote the manuscript; J.K. designed the experiments, collected data, identified species and co-wrote the manuscript; M.S. collected data and co-wrote the manuscript; M.Z. collected data, identified species and co-wrote the manuscript; and P.Z. collected data, performed statistical data processing and co-wrote the manuscript. All authors have read and agreed to the published version of the manuscript.

**Funding:** This research was funded by the Slovak Research and Development Agency (APVV), via grant nos. APVV-0707-12, APVV-15-0348, APVV-14-0567 and APVV-15-0531, as well as by the project "SLOVLES", from the Ministry of Agriculture and Rural Development of the Slovak Republic. The research was also supported by the Slovak Grant Agency for Science (VEGA) via grant nos. VEGA 2/0012/17 and VEGA 2/0032/19.

**Acknowledgments:** We thank two anonymous reviewers for their valuable comments to the former manuscript. We thank Dominique Fournier (Canada) for linguistic and editorial improvements and Milan Mikuš for technical assistance with data collecting.

**Conflicts of Interest:** The authors declare no conflict of interest.

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
