# Peer review of "Winter Geometrid Moths in Oak Forests: Is Monitoring a Single Species Reliable to Predict Defoliation Risk?"

_forests, doi:10.3390/f11030288_

Round 1

Reviewer 1 Report

The paper provides additional knowledge to geometrid moth research by analyzing the abundance of spring species, autumn species, caterpillars and secondary data. It confirms certain findings in previous studies.

The types of moths under study are not clearly elaborated in the first sentence of the abstract. Make clear what winter moths, spring species and autumn species mean. In addition show the relationship between moths and caterpillars.

Larvae/ larval stages are mentioned in the abstract, introduction, methods and discussion but no data shown or described in the results section.

The abstract does not highlight the objective or hypothesis of the study.

Line 48: “these species” not clear which species are you referring to

Line 53: not clear revise sentence

Line 59: including ‘them”

Line 65: trunks “of” 45 adult trees not “from”

Line 68:  Not clear, describe what was characteristics were determined from the collected samples

Line 71: Revise sentence, it is not clear “these were other……..” which trees?

How did you sample the trees (45 and 23); what was the size of area of study

Line 79: which areas are you referring to?

Line 80: which several years are you referring to?

Lines 89-93: does not belong to statistical section; rather move this text to the section on collection methods

Line 104: revise to “statistical analysis”

Line 116: add bar charts to show males distribution.

In addition, show the larval data / bar charts that were described in the methods section in lines 74-77.

Lines 123-130: add a summary table for this section

Lines 134 – 143: add a summary table for this section

Your data is not enough show species distributions counts means and standard deviations, statistical significance. You cannot make conclusions based on just Figure 1 plus secondary data. The conclusions you make need to be consistent with the arguments stated and supporting data.

Lines 209-210: There is no data on defoliation therefore the title should not speak of defoliation as it can be misleading.

Reviewer 2 Report

This is an interesting study and the authors have collected good data using traditional methods in forest entomology.  

The authors show how the number of winter moth adults recorded only in late autumn and/or early winter does not reflect the one of all caterpillars belonging to this group of geometrids over the following spring.

The authors recommend adults recording should be continued up to early spring to include ‘spring species’ as well. This approach can even improve data about ‘autumn species.’  

I advise to reorganize the results section in such a way to make them easily understandable for the readers. As a consequence, the discussion section needs to be revised too.  

Overall, I suggest to re-write the results and discussion sections.

Best regards
